# General versus Brachial Plexus Block Anesthesia in Pain Management after Internal Fixation in Patients with Distal Radius Fracture: A Randomized Controlled Trial

**DOI:** 10.3390/ijerph19159155

**Published:** 2022-07-27

**Authors:** Jae-Hwi Nho, Byung-Woong Jang, Chi Young An, Jae Hwa Yoo, Sanghoon Song, Ho Bum Cho, Sang Ho Kim, Soon Im Kim, Ki Jin Jung, Byungsung Kim

**Affiliations:** 1Department of Orthopaedic Surgery, Soonchunhyang University Hospital Seoul, Seoul 04401, Korea; huuy@schmc.ac.kr (J.-H.N.); 132523@schmc.ac.kr (C.Y.A.); 2Department of Orthopaedic Surgery, Soonchunhyang University Hospital Gumi, Gumi 39371, Korea; 3Department of Anesthesiology and Pain Medicine, Soonchunhyang University Hospital Seoul, Seoul 04401, Korea; 89523@schmc.ac.kr (J.H.Y.); 112226@schmc.ac.kr (S.S.); c108766@schmc.ac.kr (H.B.C.); pc@schmc.ac.kr (S.H.K.); soonnim@schmc.ac.kr (S.I.K.); 4Department of Orthopaedic Surgery, Soonchunhyang University Hospital Cheonan, Cheonan 31151, Korea; c89546@schmc.ac.kr; 5Department of Orthopaedic Surgery, Soonchunhyang University Hospital Bucheon, Bucheon 14584, Korea; kbsos@schmc.ac.kr

**Keywords:** distal radius fracture, postoperative pain management, brachial plexus block, general anesthesia

## Abstract

Distal radius fractures (DRFs) are very common injuries associated with aging, and the number of fractures is increasing with the increase in the elderly population. General anesthesia or brachial plexus block (BPB) is required for fracture fixation, and acute postoperative pain control is necessary after operation. Early pain control can improve patient satisfaction and functional outcomes. In this study, we report the clinical differences in postoperative pain, according to the method of anesthesia (general anesthesia versus brachial plexus block). Volar plating was used to treat 72 patients older than 60 years who had comminuted DRF. Patients were randomized to either group A (36 patients), who underwent general anesthesia, or group B (36 patients), who underwent BPB. We compared these two groups prospectively for acute postoperative pain using a visual analog scale (VAS) at 2, 4, 6, 12, and 24 h after surgery. The VAS scores of each group were: 6.8 ± 2.5 in general anesthesia and 0.5 ± 2.3 in BPB at 2 h, postoperatively; 6.5 ± 2.4 in general anesthesia and 0.5 ± 2.4 in BPB anesthesia at 4 h, postoperatively; 5.2 ± 2.4 in general anesthesia and 1.5 ± 2.4 in BPB anesthesia at 6 h, postoperatively; 4.5 ± 2.5 in general anesthesia and 3.4 ± 2.7 in BPB anesthesia at 12 h, postoperatively; and 3.5 ± 2.5 in general anesthesia and 3.2 ± 2.7 in BPB anesthesia at 24 h, postoperatively. DRF patients with BPB anesthesia showed a lower VAS score than those subjected to general anesthesia in early postoperative period. As a result, the effect of BPB anesthesia on acute pain management after surgery was excellent, which resulted in a lower pain score compared with general anesthesia in DRF patients undergoing volar plating.

## 1. Introduction

Distal radius fractures (DRFs) are very common injuries, and the number of fractures is increasing with the increase in the elderly population, as the density of bone decreases with age [1,2]. Several surgical methods are used to treat DRFs including open reduction and internal fixation (ORIF) with plates or screws, external fixation, and percutaneous fixation. The selection of the correct approach is still disputed by surgeons. However, anatomic reduction via stable volar plate fixation is the preferred treatment for DRF, because it leads to early function recovery after surgery [3,4].

Typical methods of anesthesia used to perform surgery for DRFs include general anesthesia (GA) and brachial plexus block (BPB). BPB is superior to GA in decreasing postoperative nausea, vomiting, reducing the need for perioperative opiate consumption, and shortening the stay in the postoperative care unit [2,5,6]. BPB also leads to control of pain after surgery for GA. Early postoperative pain control can improve patient satisfaction and improve functional outcomes [5,6].

The advantages and benefits of BPB compared with GA have been reported previously [7]. However, few comparative studies reported pain management in the acute phase after DRF surgery. The purpose of this study is to investigate the postoperative pain control and clinical differences in patients undergoing DRF operation with BPB compared with GA.

## 2. Materials and Methods

### 2.1. Patient Selection

This is a case-control study conducted with prospectively applied data and approved by the institutional review board. We included 72 patients who underwent operation for DRFs from November 2019 to April 2021. The inclusion criteria were: (1) female sex, (2) 60 years of age and over, (3) patients who did not have an underlying life-threatening disease (ASA 1 or ASA 2), (4) no cognitive impairment, and (5) understanding of the VAS score before operation.

A randomized number was generated using an automated computer program. Randomization was performed when the patient arrived in the operating room using randomly numbered and sealed envelopes. The envelopes were placed in the operating room and only opened when the patient arrived, and neither the patient nor the involved physicians were aware of the randomization. Patients who refused randomization or already preferred one of the anesthetic techniques were excluded from the study. Finally, patients were randomized to either group A (36 patients), who underwent general anesthesia, or group B (36 patients), who underwent BPB anesthesia.

The average age of 36 patients who underwent operation under GA (Group A) was 71.2 ± 14.5 years. The average age of another 36 patients who were treated surgically under BPB (Group B) was 69.5 ± 15.8 years. The right wrist was injured in 14 patients and the left wrist was injured in 22 patients in Group A, and the right wrist was injured in 22 patients and the left wrist was injured in 14 patients belonging to Group B (Table 1).

### 2.2. Methods of Anesthesia, Operation, and Pain Assessment

In Group A, induction was performed using intravenous lidocaine 40 mg, fentanyl 0.5 mg/kg, propofol 1 to 1.5 mg/kg, and rocuronium 0.6 mg/kg for neuromuscular blockade. A supraglottic airway was inserted based on patient size. Anesthesia was maintained with oxygen, medical air, and 1 minimum alveolar concentration of desflurane under monitoring of bispectral index (40 to 60). During skin closure, neuromuscular blockade was reversed with 0.2 mg/kg pyridostigmine and 5 mg/kg glycopyrrolate [8].

Ultrasound-guided axillary BPB was performed in-plane with a 25 gauge 6 cm plane needle. An anesthetics mixture was prepared including 20 mL of 0.45% ropivacaine and 3 mL of 2% lidocaine with 5 mg of dexamethasone as an adjuvant. All axillary block approaches require the patient to be positioned supine, with the arm abducted 90 degrees and the head turned toward the contralateral side. The axillary artery pulse was palpated, and its location was marked as a reference point. Using a high-frequency linear array ultrasound transducer, the axillary artery and vein were visualized in cross-section. The brachial plexus was identified surrounding the artery. The needle was inserted superior (lateral) to the transducer and advanced inferiorly (medially) toward the plexus under direct visualization. Local anesthetic mixture was then injected around each nerve. The effect of BPB was evaluated with a temperature test using an alcohol swab; then, sedation was performed using 2% propofol 50 μg/kg/min after checking anesthesia status [9].

A single surgeon (J-H N) performed volar plating in all the investigated patients. The volar approach (Classical Henry approach) was used along the flexor carpi radialis tendon. Fractures were fixed using a volar locking plate system (Acumed, Hillsboro, OR, USA) after achieving anatomical reduction. Following volar plating, the PQ muscle was repaired via insertion as close as possible using absorbable sutures with the forearm in supination at the end of the operation. Postoperatively, the wrists were immobilized using a short-arm volar plaster splint.

After the operation, all patients were evaluated for acute postoperative pain based on the VAS score at 2, 4, 6, 12, and 24 h after the operation. We evaluated the sensory recovery over time after surgery in group B as follows: (1) time of restoration of sensory awareness of patients; (2) time of pain initiation; and (3) time when patients’ senses recovered fully without any numbness or hypoesthesia on affected hand. A self-report questionnaire form including these items was handed out to the patients before the surgery. The patient was allowed to fill the questionnaire after the surgery, and the physician confirmed it.

Postoperative pain medications were designed based on our previous experiences and currently available literature on acute pain management. Patients were prescribed a nonsteroidal anti-inflammatory drug (NSAID); celecoxib 200 mg orally twice per day. Rescue medication was prescribed through intravenous short-acting narcotics; tramadol 50 mg to 100 mg every 4 to 6 h. There were no differences between groups.

### 2.3. Statistical Analysis

A power analysis was performed using the software package G power (version 3.1.9.4.) (Faul and Erdfelder, 1992) [10]. Power analysis calculations were based on an 80% power to detect a difference in VAS scale between both groups (One-tail *t*-test; a = 0.05). The analysis indicated that a sample size of 35 patients per group was required.

Statistical analysis was carried out using SPSS (version 21.0; IBM, Armonk, NY, USA). The independent *t*-test was used to compare continuous variables. The significance of the VAS score at 2, 4, 6, 12 and 24 h in each group was determined via paired *t*-test. A chi-square test was used to compare categorical variables. A significant level of *p* < 0.05 was used for all comparisons. The results were presented as mean ± standard and as number for discrete variables.

## 3. Results

### 3.1. VAS Scores after Operation

The VAS scores after operation of each group were as follows: 6.8 ± 2.5 in Group A and 0.5 ± 2.3 in Group B at 2 h postoperatively; 6.5 ± 2.4 in Group A and 0.5 ± 2.4 in Group B at 4 h postoperatively; 5.2 ± 2.4 in Group A and 1.5 ± 2.4 in Group B at 6 h postoperatively; 4.5 ± 2.5 in Group A and 3.4 ± 2.7 in Group B at 12 h postoperatively; and 3.5 ± 2.5 in Group A and 3.2 ± 2.7 in Group B at 24 h postoperatively (Table 2). The pain levels of both groups converged to similar levels about 24 h after surgery (Figure 1).

### 3.2. Sensory Recovery over Time after Surgery with BPB Anesthesia

The results of sensory recovery over time after surgery in group B indicated that (1) the average time required to restore patients’ senses was 10.5 ± 2.8 h; (2) patients’ pain started in 12.4 ± 3.9 h; and (3) the average time required for complete recovery of patients’ senses was 19.2 ± 4.1 h (Table 3).

## 4. Discussion

Many patients experience strong pain after volar plate fixation of displaced radius fractures [11,12]. DRF surgery has been performed under several modes of anesthesia such as general and regional anesthesia, and even under wide-awake local anesthesia no tourniquet (WALANT) technique [13]. BPB is widely used because of its safety and excellent pain control in extremity surgery. Acute pain management after surgery has several clinical implications. Reducing the level of pain after surgery allows the patient to participate more actively in rehabilitation, resulting in faster recovery [14]. Furthermore, effective postoperative pain control is an essential component of care of the surgical patient, because inadequate pain control results in increased morbidity or mortality [15]. We investigated the pattern of the acute phase of postoperative pain and clinical differences among patients who underwent DRF operation with BPB compared with GA. In this study, we investigated 72 female patients who underwent DRF surgery with volar plate. There is a difference in the degree of pain adjustment according to sex and age. Therefore, to reduce this bias, only females over 60 years of age with postmenopausal osteoporotic fracture were the subject of this study.

The followup results of the pain score up to 24 h postoperatively revealed that patients in the BPB group suffered less pain than those in the GA group at all time points. Therefore, faster rehabilitation and recovery can be expected via effective pain control using BPB. However, rebound pain is a disadvantage of BPB and has been reported in several studies. Galos et al. [11] reported the results of a randomized study showing a higher pain score in the BPB group between 6 and 48 h postoperatively due to rebound pain. In this study, the pain score in the BPB group increased rapidly from 1.5 to 3.4 between 6 and 12 h postoperatively, but the pain score was still lower than in the GA group. Ropivacaine and bupivacaine are long acting-agents, and their durations of action are similar, ranging from 90 to 480 min. Galos et al. injected 15 cc of 0.25% bupivacaine along with lidocaine, whereas in this study, higher doses of ropivacaine (20 cc of 0.45%) and 5 mg of adjuvant dexamethasone were used. This difference in anesthetic mixture is thought to have reduced rebound pain after BPB anesthesia in this study. Holmberg et al. reported that intravenous dexamethasone improves early postoperative analgesia and may also improve clinical outcomes after 6 and 12 months [16].

BPB can be performed at different anatomical sites such as interscalene, supraclavicular, infraclavicular, axillary nerve regions, and the humeral canal [17,18]. In addition, the brachial plexus is adjacent to important anatomical structures and increases the risk of catastrophic complications related to needle puncture such as pneumothorax, nerve palsy, and hematoma [19,20]. Therefore, BPB requires careful preparation and great attention. Axillary nerve block and humeral canal block can be used for surgery at the elbow or below elbow level, with similar efficacy and low complication. However, the axillary nerve block may be associated with decreased time related to block performance and onset [21].

One of the strengths of this study was the monitoring of sensory changes in the BPB group over time. The mean time required to restore patients’ senses was 10.5 ± 2.8 h, and the mean time need to trigger pain was 12.4 ± 3.9 h. This finding can be interpreted in relation to the result of a rapid increase in the VAS score 6–12 h after surgery in patients subjected to BPB. Therefore, we suggest that to control this pain pattern after DRF surgery, an appropriate pain reliever should be used approximately 12 h after surgery, when rebound pain is triggered.

Patients with fractures involving the upper extremity are at high risk for developing complex regional pain syndrome type I (CRPS I) [22], with reported incidence rates after a DRF ranging from 1 to 37% [23,24,25,26]. One possible explanation for the improved outcomes in patients receiving regional anesthesia is the concomitant blockade of the sympathetic nervous system at the time of surgery. There are many case reports in which the CRPS I symptoms were successfully controlled via BPB [27,28]. In this study, we focused only on the pain control during the acute phase after surgery; however, sympathetic blockade may result in a lower rate of CRPS or at the very least act as an early intervention to interrupt the course of a developing pain syndrome.

This study had some limitations. First, we investigated patients’ pain based on VAS, which is a subjective score. It was a challenge to mask the surgeon, anesthesiologist, and patients continuously after the operation. In addition, only the results of the short-term pain scale up to 24 h after surgery were compared in this study. A study including long-term clinical outcomes and functional scores is required to confirm the effect of BPB over GA.

## 5. Conclusions

The effect of BPB anesthesia on acute pain management after surgery was excellent, which resulted in a lower pain score compared with general anesthesia in DRF patients undergoing volar plating. In addition with our local anesthetic regimen of BPB anesthesia, we could expect to reduce the rebound pain of BPB.

## Figures and Tables

**Figure 1 ijerph-19-09155-f001:**
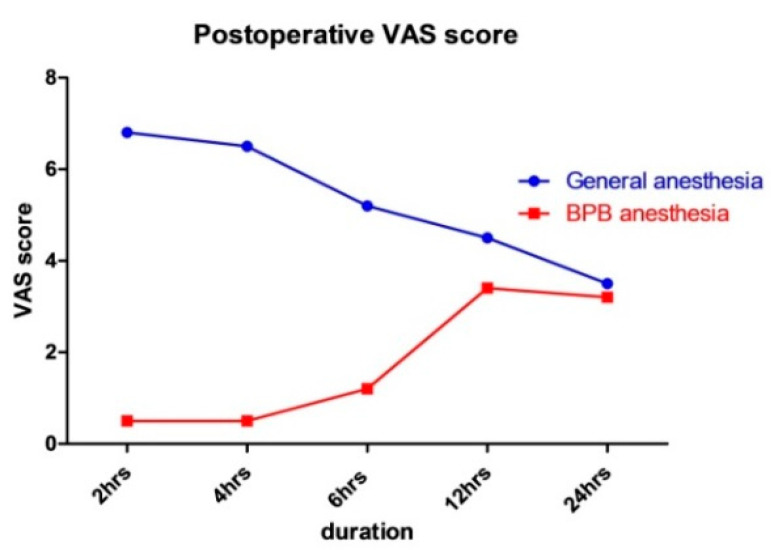
VAS scores over time after DRF surgery in the GA and BPB groups.

**Table 1 ijerph-19-09155-t001:** Demographic data of patients treated with volar plating for distal radius fracture.

		General Anesthesia (Group A)	BPB Anesthesia (Group B)	*p*-Value
Age (mean)		71.2 ± 14.5	69.5 ± 15.8	0.687
Injured wrist	Right	14	22	0.061
	Left	22	14	
Fracture type	A	1	3	<0.001
	B	0	0	
	C1	26	19	
	C2	8	13	
	C3	1	1	
Operation time (mean, minute)		31.9 ± 5.2	34.0 ± 2.7	<0.001

Note. M = male, F = female, BPB = brachial plexus block.

**Table 2 ijerph-19-09155-t002:** Visual analog scale (VAS) scores in both groups at different times after surgery.

Time after Surgery	General Anesthesia (Group A)	BPB Anesthesia (Group B)	*p*-Value
2 h	6.8 ± 2.5	0.5 ± 2.3	<0.001
4 h	6.5 ± 2.4	0.5 ± 2.4	<0.001
6 h	5.2 ± 2.4	1.5 ± 2.4	<0.001
12 h	4.5 ± 2.5	3.4 ± 2.7	<0.001
24 h	3.5 ± 2.5	3.2 ± 2.2	0.142

Note: BPB = brachial plexus block.

**Table 3 ijerph-19-09155-t003:** Results of sensory recovery over time after surgery in Group B.

	Average Time (h)
Senses begin to return	10.5 ± 2.8
Pain begins	12.4 ± 3.9
Complete recovery of senses	19.2 ± 4.1

## Data Availability

The data presented in this study are available on request from the corresponding author. The data are not publicly available due to patient privacy.

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
