# Peer review of "General versus Brachial Plexus Block Anesthesia in Pain Management after Internal Fixation in Patients with Distal Radius Fracture: A Randomized Controlled Trial"

_ijerph, 2022, doi:10.3390/ijerph19159155_

Round 1

Reviewer 1 Report

Congratulations to the authors. This manuscript is clearly written and designed. The authors show that there is a statistically significant difference between VAS Scores for patient receiving BPB versus general anesthesia for DRF ORIF procedures. The  limitations of the study are also noted by the authors. I am wondering if other pain measurement modalities or questionnaires were used to assess postoperative pain, also if data was obtained regarding postoperative appointments multiple months after surgery and if this patient population was followed for a longer period of time after surgery. I am also wondering if a power calculation was performed as there were 72 patients included. Overall I believe the study is clearly designed and written. 

Reviewer 2 Report

This prospective randomized trial studied clinical differences in postoperative pain intensity in patients undergoing volar plating for distal radius fracture between a group of 72 patients older than 60 years  who underwent the procedure under general anesthesia (Group A, n=36) versus a group who received a brachial plexus block plus sedation (Group B, n=36). Authors based their comparison for effectiveness of any anesthesia technique in the measured postoperative pain intensity by VAS scale, and sensory recovery. Although this manuscript addresses an important problem that the anesthesiologist must solve, such as the management of postoperative pain in orthopedic surgery by methods that reduce the consumption of opioids, it presents a significant number of flaws that require extensive modifications. Below I present my comments.

Major Comments:

Title

The title dose not describe the main outcomes of the study.

Abstract

1)      Page 1, line 17: delete repeated “improve” or change it for a synonym.

Material and Methods

There are several major methodological flaws that I detected:

1)      The authors must explain why, being distal radius fractures a frequent acute pathology in both sexes and in all age groups, they only chose female patients and patients older than 60 years.

2)      In the demographic data summarized in Table 1 there is a disproportionated difference between the operation time in Group A (31.9 minutes) and Group B (3.4 ± 2.7 minutes), please explain the reason for this bias.

3)      The description of the technique chosen for performing the BPB should be more detailed including: anatomical approach (supraclavicular, interscalene, infraclavicular), patient position, anesthetic mixtures, type of U/S probe, location of the neural or perineural injection site with ultrasound in relation to the anatomical structures, safety measures during the block to avoid inadvertent intravascular injection, etc. It is noteworthy that the surgical technique is better described than the regional anesthetic technique. Also, the description of the anesthetic mixture need a better wording.

4)      Please explain the meaning of “the time when patient’s sense are recovered fully” (Page 3, lines 103-104). The sentence is very vague.

5)      There is no mention of postoperative analgesic regime as well as rescue medication used in both groups.

Statistical Analysis

1)      The statistical analysis should mention whether continuous variables were measured in means or medians.

Results

1)      In Table 2 all p-values should be revised.

2)      Please define more exactly the term sensory awareness (page 4, lines 128-129).

3)      Explain if the data for sensory monitoring was also collected for Group A and if it was collected because it was not included in Table 3 to compare the results with Group B (mean time & p-values).

Discussion

1)      Authors stated that they “investigated the patterns of acute phase postoperative pain and clinical differences among patients who underwent DRF with BPB compared with GA”. However, the results do not show data regarding sensory recovery in the group A (general anesthesia). Therefore, I conclude that there is no evidence for such comparison (page 5, lines 141-142).

2)      According to authors: “the pain score was still lower than in the GA group, which may be due to differences in the method of BPB or drugs”. Please mention to which drugs they refer to (page 5, lines 151-153).

3)      The statement:” The axillary nerve block yielded good results when performed by an orthopedic surgeon, and not an anesthesiologist” (page 5, lines 167-168), does not add any additional value to the discussion unless some of the BPB were performed by orthopedic surgeon, which was mentioned in the description of the methodology.

Conclusion

The conclusion needs to be revised in order to improve the wording and spelling for a better understanding.

Minor Comments

1)      In the Abstract portion: Delete repeated “improve” or substitute it for a synonym (page 1, line 17).

Round 2

Reviewer 2 Report

 I am glad for the effort made by the authors to improve the quality of the manuscript. and adjunct tables. However, in my opinion there are few minor changes to make.

1) In the Methods Section: 'All the numerous axillary block techniques" should be change for "all axillary block approaches" (page 3, line 87)

2) In page 3, lines 94-95: "The effect of local block was determined with a temperature 95 test using an alcohol swab". I don't have a clear idea about how the temperature test to determine the effectiveness of the block works, beside the reference provided to support it is in Chinese. 

3) In the discussion section: the explanation provided to justify the choice of the study population (female, >60 years) is still insufficient. 

Author Response

1) In the Methods Section: 'All the numerous axillary block techniques" should be change for "all axillary block approaches" (page 3, line 87)

Res: Thank you for your kind correction. As suggested, I’ve revised it.

2) In page 3, lines 94-95: "The effect of local block was determined with a temperature 95 test using an alcohol swab". I don't have a clear idea about how the temperature test to determine the effectiveness of the block works, beside the reference provided to support it is in Chinese. 

Res: Thank you for your valid point. I’ve revised the text more clearly.

“The effect of BPB was evaluated with a temperature test using an alcohol swab, then sedation was performed using 2% propofol 50μg/kg/min after checking anesthesia status.”

3) In the discussion section: the explanation provided to justify the choice of the study population (female, >60 years) is still insufficient. 

Res: I understand your concerns. These criteria were developed to target patients with postmenopausal osteoporotic fractures. Thank you for your kind consideration. I’ve revised it clearly with your recommendation.

“Therefore, to reduce this bias, only females over 60 years of age with postmenopausal osteoporotic fracture were the subject of this study.”